# High-dimensional Bayesian Optimization via Semi-supervised Learning with Optimized Unlabeled Data Sampling

## Abstract

Bayesian optimization (BO) is a powerful sequential optimization approach for seeking the global optimum of black-box functions for sample efficiency purposes. Evaluations of black-box functions can be expensive, rendering reduced use of labeled data desirable. For the first time, we introduce a teacher-student model, called `TSBO`, to enable semi-supervised learning that can make use of large amounts of cheaply generated unlabeled data under the context of BO to enhance the generalization of data query models. Our teacher-student model is uncertainty-aware and offers a practical mechanism for leveraging the pseudo labels generated for unlabeled data while dealing with the involved risk. We show that the selection of unlabeled data is key to `TSBO`. We optimize unlabeled data sampling by generating unlabeled data from a dynamically fitted extreme value distribution or a parameterized sampling distribution learned by minimizing the student feedback. `TSBO` is capable of operating in a learned latent space with reduced dimensionality, providing scalability to high-dimensional problems. `TSBO` demonstrates the significant sample efficiency in several global optimization tasks under tight labeled data budgets.

## 1 Introduction

Many science and engineering tasks such as drug discovery (Dai et al., 2018; Griffiths & Hernández-Lobato, 2020), structural design and optimization (Zoph et al., 2018; Ying et al., 2019; Lukasik et al., 2022), and failure analysis (Hu et al., 2018; Liang, 2019) can be formulated as a black-box function optimization problem in a given input space, which can be high-dimensional. Despite the remarkable progress achieved in recent years through the application of advanced machine learning techniques to various optimization problems, addressing the issue of developing global optimization methods for problems characterized by agnostic objective functions, all while minimizing the number of function evaluations, remains a significant and ongoing challenge.

Bayesian Optimization (BO) stands as a sequential and sample-efficient methodology employed in the quest for global optima within black-box functions (Brochu et al., 2010; Snoek et al., 2012). BO comprises two main components: a surrogate probabilistic model for regressing the objective function with a posterior predictive distribution, and an acquisition function for new data query while trading off between exploration and exploitation. A typical iteration of BO unfolds in three stages: 1) a surrogate, typically a Gaussian Process (GP) (Seeger, 2004), is fitted on labeled data; 2) an acquisition function based on the posterior predictive distribution is optimized to pick the next query point; 3) The new queried data pair is added to the training dataset for the next BO iteration. Given the evaluation of the black-box function is potentially exorbitant, it is desirable to reduce the use of expensive labeled data.

Semi-supervised learning offers a promising avenue for mitigating the challenge of limited labeled data by harnessing abundant, inexpensive unlabeled data. In the context of high-dimensional BO, it is a widely adopted practice to utilize unlabeled data[1] to learn 1) a dimension reduction model to induce a low-dimensional latent space where BO is performed, and 2) a generative model to project

---

[1]Although the term label often appears in classification problems, it is also widely used to represent observed values in BO (Grosnit et al., 2021; Chen et al., 2020; Jean et al., 2018).

the latent code with the optimum acquisition value to the original space for evaluation. Recent work has explored various encoding and decoding models, including linear projection (Chen et al., 2020), nonlinear embedding (Moriconi et al., 2020), and Variational Autoencoder (VAE) (Kusner et al., 2017; Jin et al., 2018; Tripp et al., 2020; Grosnit et al., 2021). However, the utilization of unlabeled data for GP data query model training remains uncharted territory, to the best of our knowledge. The primary challenge stems from the fact that GPs inherently rely on labeled data and cannot directly accommodate data lacking label information.

To address this challenge, we propose a novel approach that involves the direct incorporation of unlabeled data into Gaussian Process (GP) data query models by leveraging pseudo-label predictions, thereby enhancing model generalization. Our technique is compatible with the aforementioned semi-supervised latent space BO methods. Our main contributions are:

• We present Teacher-Student Bayesian Optimization (`TSBO`), a semi-supervised learning BO method with a novel pseudo-label dependent uncertainty-aware teacher-student model.

• We systematically optimize the locations of unlabeled data by sampling from 1) a dynamically fitted extreme value distribution, or 2) a parameterized sampling distribution learned by minimizing the student's feedback loss.

• We empirically demonstrate the significant sample efficiency of `TSBO` in high-dimensional global optimization tasks. In a chemical design task, `TSBO` improves a recent latent space BO approach to achieve a similar molecular profile score within 3‰ total evaluations.

## 2 PRELIMINARIES

**BO Objective** Given a set of $N$ sequentially queried labeled examples $\{\mathbf{x}_i, y_i\}_{i=1}^N = \{\mathbf{X}_l, \mathbf{y}_l\}$, where the $N \times D$ matrix $\mathbf{X}_l$ and the $N \times 1$ vector $\mathbf{y}_l$ are the inputs and the corresponding observed target values (labels), respectively, we aim to solve an optimization problem:

$$\mathbf{x}^* = \operatorname*{argmax}_{\mathbf{x} \in \mathcal{X}} f(\mathbf{x}), \tag{1}$$

where $\mathcal{X} \subseteq \mathbb{R}^D$ is a $D$-dimensional input space, and $f : \mathcal{X} \to \mathbb{R}$ is an agnostic function. Under BO, we desire to find the global maximum $\mathbf{x}^*$ with a minimal amount of expensive data query.

**Latent Space BO** When the dimension $D$ of $\mathcal{X}$ scales to a large value, applying BO will suffer from the curse of dimensionality (Brochu et al., 2010). An effective solution to this challenge lies in the realm of latent space BO Kusner et al. (2017), where BO is deployed in a low-dimensional latent space $\mathcal{Z} \subseteq \mathbb{R}^d$ such that $d \ll D$. With the facilitation of an encoder $\psi : \mathcal{X} \to \mathcal{Z}$ and a decoder $\varphi : \mathcal{Z} \to \mathcal{X}$, latent space BO is able to 1) fit a data query GP model on a labeled dataset $\mathcal{D}_l := \{\{\mathbf{z}_{l_i}, y_{l_i}\}\}_{i=1}^N = \{\mathbf{Z}_l, \mathbf{y}_l\}$ where $\mathbf{z}_{l_i} := \psi(\mathbf{x}_{l_i})$, 2) optimize an acquisition function to pick the optimum latent code $\tilde{\mathbf{z}}$, and 3) make a new evaluation $\tilde{y} = f(\tilde{\mathbf{x}})$ where $\tilde{\mathbf{x}} = \varphi(\tilde{\mathbf{z}})$. This latent space BO framework is visually depicted in the left part of Fig. 1.

## 3 OVERVIEW OF THE `TSBO` FRAMEWORK

To address the challenge of lack of expensive labeled data, we propose `TSBO`, a unified BO approach incorporating a teacher-student model based on semi-supervised learning. As illustrated in the right segment of Figure 1, our method involves the generation of pseudo labels for sampled unlabeled data, which are then combined with the labeled data to inform a GP model for data query. `TSBO` contains three key components:

**Pseudo-Label Dependent Uncertainty-Aware Teacher-student Model** `TSBO` employs a teacher-student model for pseudo-label prediction. The teacher-student model is optimized by solving a bi-level minimization problem, as discussed in Algorithm 1. The teacher generates pseudo labels for a set of unlabeled data, which are used to train the student and a new data query model.

To enhance the quality of pseudo labels, we further propose a pseudo-label dependent uncertainty-aware teacher-student model, integrating prediction uncertainties into the training scheme of the teacher-student. It comprises a probabilistic teacher and an uncertainty-aware student, as detailed in Section 4.

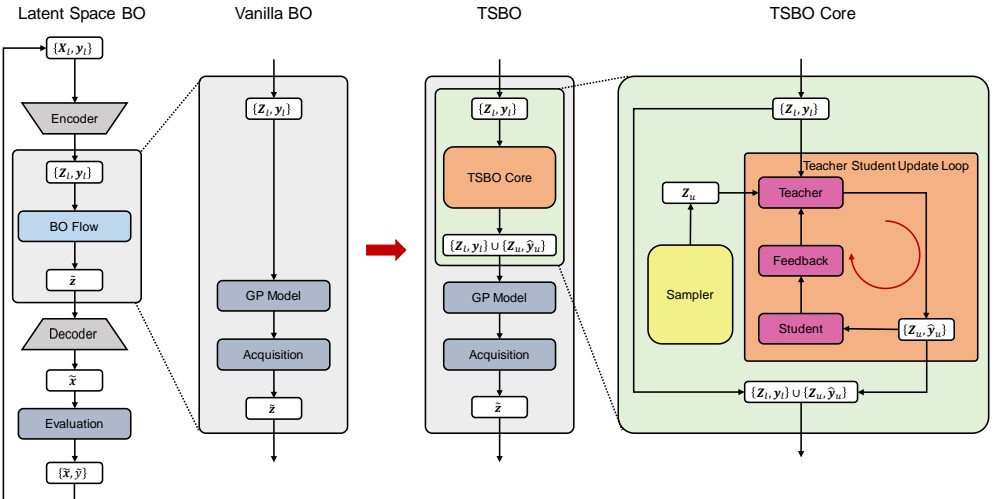

Figure 1: A comparison between vanilla BO and `TSBO`. On the left side of the red arrow, we present the typical latent space BO workflow. on the right side, we provide the overview of `TSBO`, involving a teacher-student model for pseudo-label prediction, and an optimized unlabeled data sampling strategy.

**Unlabeled Data Samplers** In each BO iteration, in addition to the current labeled dataset, we dynamically generate an optimized unlabeled dataset. We find that the selection of unlabeled data has a significant impact on the final performance. The proposed approach is discussed in Section 5.

**Data Query GP Model** Our data query GP fits on the combination of the labeled and unlabeled data, along with their associated pseudo labels. The generalization of our GP is enhanced by our pseudo-label prediction technique.

## 4 DESIGN OF UNCERTAINTY-AWARE TEACHER-STUDENT MODEL

We propose a pseudo-label dependent uncertainty-aware teacher-student model, where the teacher is probabilistic and outputs both the mean and variance of a pseudo label for an unlabeled sample. Furthermore, our GP-based student model is *uncertainty-aware*, i.e., it fits the unlabeled dataset while taking the uncertainties of the pseudo labels into account.

We argue that this uncertainty-aware approach is essential for enabling teacher-student-based semi-supervised learning. On one hand, better student and data query models may be trained with abundant unlabeled data with pseudo labels, allowing wider exploration of the input space towards finding the global optimum without needing additional expensive labeled data. On the other hand, poorly predicted pseudo labels can jeopardize the above modeling process and result in misleading feedback to the teacher. As such, the variance of each pseudo label predicted by a probabilistic teacher can serve as a measure of the teacher's uncertainty (confidence). It is important to note that the teacher's uncertainty varies from pseudo label to pseudo label. The student shall be informed of the teacher's uncertainty and judiciously make use of a pseudo label in a way appropriate to its uncertainty level. Our uncertainty-aware teacher-student model offers a practical mechanism for leveraging pseudo labels while mitigating the involved risk.

### 4.1 PROBABILISTIC TEACHER MODEL

`TSBO` employs a probabilistic teacher $T$, which is a multilayer perceptron (MLP) parameterized by $\boldsymbol{\theta_T}$. For a given latent input $\mathbf{z} \in \mathcal{Z}$, the teacher's output $T(\mathbf{z}; \boldsymbol{\theta_T})$ is considered to follow a Gaussian distribution $T(\mathbf{z}; \boldsymbol{\theta_T}) \sim \mathcal{N}(\mu_{\boldsymbol{\theta_T}}(\mathbf{z}), \sigma^2_{\boldsymbol{\theta_T}}(\mathbf{z}))$, where $\mu_{\boldsymbol{\theta_T}}(\mathbf{z}) \in \mathbb{R}$ and $\sigma^2_{\boldsymbol{\theta_T}}(\mathbf{z}) \in (0, +\infty)$ are the predicted mean and its variance from the teacher model. The training of the teacher is described in Section 4.4.

---

**Algorithm 1** Bi-Level Optimization of the Teacher-Student Model

---

**Input:** Epochs $L$, feedback weight $\lambda$, teacher $T(\cdot; \boldsymbol{\theta}_T^0)$, student $S(\cdot; \boldsymbol{\theta}_S^0)$, labeled data $\mathcal{D}_l$, validation data $\mathcal{D}_u$, unlabeled data $\mathbf{Z}_u$
**Output:** Pseudo labels $\hat{\mathbf{y}}_u$
**for** $i = 1$ to $L$ **do**
    Generate pseudo labels: $\hat{\mathbf{y}}_u \leftarrow \mathbb{E}\left(T(\mathbf{Z}_u; \boldsymbol{\theta}_T^{i-1})\right)$
    Update the student model: $\boldsymbol{\theta}_S^i \leftarrow \boldsymbol{\theta}_S^{i-1} - \eta_S \cdot \nabla_{\boldsymbol{\theta}_S^{i-1}} \mathcal{L}_u(\mathbf{Z}_u, \boldsymbol{\theta}_T^{i-1}; \boldsymbol{\theta}_S^{i-1})$ via Eq. (2)
    Fix $\boldsymbol{\theta}_S^i$, and update the teacher model: $\boldsymbol{\theta}_T^i \leftarrow \boldsymbol{\theta}_T^{i-1} - \eta_T \cdot \nabla_{\boldsymbol{\theta}_T^{i-1}}\{\lambda \mathcal{L}_f(\mathbf{Z}_v, \mathbf{y}_v; \boldsymbol{\theta}_S^i, \boldsymbol{\theta}_T^{i-1}, \mathbf{Z}_u) +$
    $\mathcal{L}_l(\mathbf{Z}_l, \mathbf{y}_l; \boldsymbol{\theta}_T^{i-1})\}$ via Eq. (5) and Eq. (7)
**end for**
Predict pseudo labels: $\hat{\mathbf{y}}_u \leftarrow \mathbb{E}\left(\mu_{\boldsymbol{\theta}_T}(\mathbf{Z}_u; \boldsymbol{\theta}_T^L)\right)$

---

## 4.2 UNCERTAINTY-AWARE STUDENT MODEL

The student model $S$ of TSBO is a GP, whose learnable hyperparameters $\boldsymbol{\theta}_S$ consist of a prior mean value $\mu_0 \in \mathbb{R}$, an observation noise variance value $\sigma_0^2 > 0$, a scalar parameter $\tau$, and a lengthscale $l$ of a Radial Basis Function (RBF) kernel: $\kappa_0(\mathbf{z}, \mathbf{z}') = \tau^2 \exp(-\|\mathbf{z} - \mathbf{z}'\|^2/2l^2)$, which governs the correlations between points in the latent space $\mathcal{Z}$ (Rasmussen & Williams, 2006).

Our student $S$ is optimized over the unlabeled dataset $\mathcal{D}_u := \{\{\mathbf{z}_{u_i}, \hat{y}_{u_i}\}\}_{i=1}^M = \{\mathbf{Z}_u, \hat{\mathbf{y}}_u\}$, where each pseudo label $\hat{y}_u$ is set to the corresponding mean prediction $\mu_{\boldsymbol{\theta}_T}(\mathbf{z}_u)$ from the teacher $T$: $\hat{y}_u = \mu_{\boldsymbol{\theta}_T}(\mathbf{z}_u)$, representing $T$'s best prediction of the unknown ground truth label $y_u$.

The development of our pseudo-label dependent uncertainty-aware student model $S$ involves two important treatments: 1) First, the uncertainty in the teacher's pseudo label generation process is modeled by: $\hat{y}_u = y_u + \epsilon_u(\mathbf{z}_u)$, where $\epsilon_u(\mathbf{z}_u) \sim \mathcal{N}(0, \sigma_{\boldsymbol{\theta}_T}^2(\mathbf{z}_u))$. The pseudo-label dependent noise $\epsilon_u(\mathbf{z}_u)$ is *intrinsic* to the teacher. 2) Next, we propagate the teacher's intrinsic uncertainty in terms of $\epsilon_u(\mathbf{z}_u)$ to the downstream training of the student GP model by forming a student's prior: $\hat{y}_u = y_u + \epsilon_u(\mathbf{z}_u) + \epsilon_{\kappa 0}$, where $\epsilon_{\kappa 0} \sim \mathcal{N}(0, \kappa_0(\mathbf{z}_u, \mathbf{z}_u) + \sigma_0^2)$. $\epsilon_{\kappa 0}$ is a pseudo-label independent learnable additive noise modeled after the RBF kernel $\kappa_0(\cdot, \cdot)$ and the observation noise $\sigma_0^2$. This additional noise $\epsilon_{\kappa 0}$ accounts for the error in the teacher's uncertainty estimate $\sigma_{\boldsymbol{\theta}_T}^2(\mathbf{z}_u)$.

Correspondingly, we define an $M \times M$ prior covariance matrix $\mathbf{K}$ over the unlabeled dataset $\mathcal{D}_u$ as: $\mathbf{K}_{ij} = \mathbb{E}(\hat{y}_{u_i} - \mathbb{E}\,\hat{y}_{u_i})(\hat{y}_{u_j} - \mathbb{E}\,\hat{y}_{u_j}) = \kappa_0(\mathbf{z}_{u_i}, \mathbf{z}_{u_j}) + \delta_{ij}\sigma_0^2 + \delta_{ij}\sigma_{\boldsymbol{\theta}_T}^2(\mathbf{z}_{u_i})$, where $\delta$ represents the Kronecker delta. The Negative Marginal Log-Likelihood (NMLL) of $\mathcal{D}_u$ is found to be:

$$\mathcal{L}_u(\mathbf{Z}_u, \boldsymbol{\theta}_T; \boldsymbol{\theta}_S) := \left(\mu_{\boldsymbol{\theta}_T}(\mathbf{Z}_u) - \mu_0\mathbf{1}_M\right)^T \mathbf{K}^{-1}\left(\mu_{\boldsymbol{\theta}_T}(\mathbf{Z}_u) - \mu_0\mathbf{1}_M\right) + \ln|\mathbf{K}| + \text{const}, \quad (2)$$

$$\mathbf{K} = \kappa_0(\mathbf{Z}_u, \mathbf{Z}_u) + \sigma_0^2 \boldsymbol{I}_M + \text{diag}\left(\sigma_{\boldsymbol{\theta}_T}^2(\mathbf{z}_{u_1}), ..., \sigma_{\boldsymbol{\theta}_T}^2(\mathbf{z}_{u_M})\right), \quad (3)$$

where $\mathbf{1}_M$ is an $M \times 1$ vector of all ones, $\boldsymbol{I}_M$ is the $M \times M$ identity matrix, $\mu_{\boldsymbol{\theta}_T}(\mathbf{Z}_u)$ is the vector of the mean predictions of the teacher $[\mu_{\boldsymbol{\theta}_T}(\mathbf{z}_{u_1}), \cdots, \mu_{\boldsymbol{\theta}_T}(\mathbf{z}_{u_M})]^T$, the $M \times M$ kernel matrix $\kappa_0(\mathbf{Z}_u, \mathbf{Z}_u)$ is defined as $\kappa_0(\mathbf{Z}_u, \mathbf{Z}_u)_{ij} = \kappa_0(\mathbf{z}_{u_i}, \mathbf{z}_{u_j})$ for all $i, j \in \{1, ..., M\}$. We optimize the student GP model's parameters $\boldsymbol{\theta}_S$ by minimizing the NMLL ($\mathcal{L}_u$) jointly with the teacher as described in Section 4.4.

## 4.3 STUDENT'S FEEDBACK TO TEACHER

To transmit the student feedback to the teacher, we evaluate the student model over a validation dataset $\mathcal{D}_v := \{\{z_{v_i}, y_{v_i}\}\}_{i=1}^H = \{\mathbf{Z}_v, \mathbf{y}_v\}$ (discussed in Appendix B.1), giving rise to the posterior prediction $S(\mathbf{Z}_v; \mathbf{Z}_u, \boldsymbol{\theta}_T, \boldsymbol{\theta}_S)$ for the labels of $\mathcal{D}_v$:

$$
\begin{aligned}
S(\mathbf{Z}_v; \mathbf{Z}_u, \boldsymbol{\theta}_T, \boldsymbol{\theta}_S) &\sim \mathcal{N}\left(\mu_{\boldsymbol{\theta}_S}(\mathbf{Z}_v; \mathbf{Z}_u, \boldsymbol{\theta}_T), \sigma_{\boldsymbol{\theta}_S}^2(\mathbf{Z}_v; \mathbf{Z}_u, \boldsymbol{\theta}_T)\right), \\
\mu_{\boldsymbol{\theta}_S}(\mathbf{Z}_v; \mathbf{Z}_u, \boldsymbol{\theta}_T) &= \mu_0\mathbf{1}_H + \kappa_0(\mathbf{Z}_v, \mathbf{Z}_u)^T \mathbf{K}^{-1}\left(\mu_{\boldsymbol{\theta}_T}(\mathbf{Z}_u) - \mu_0\mathbf{1}_M\right), \\
\sigma_{\boldsymbol{\theta}_S}^2(\mathbf{Z}_v; \mathbf{Z}_u, \boldsymbol{\theta}_T) &= \kappa_0(\mathbf{Z}_v, \mathbf{Z}_v) - \kappa_0(\mathbf{Z}_v, \mathbf{Z}_u)^T \mathbf{K}^{-1} \kappa_0(\mathbf{Z}_v, \mathbf{Z}_u) + \sigma_0^2 \boldsymbol{I}_H.
\end{aligned}
\quad (4)
$$

We compute the Mean Square Error (MSE) between the posterior means $\mu_{\boldsymbol{\theta}_S}(\mathbf{Z}_v; \mathbf{Z}_u, \boldsymbol{\theta}_T)$ and the true labels $\mathbf{y}_v$ as our feedback loss $\mathcal{L}_f$ [2]:

$$\mathcal{L}_f(\mathbf{Z}_v, \mathbf{y}_v; \boldsymbol{\theta}_S) := \text{MSE}\Big(\mu_{\boldsymbol{\theta}_S}(\mathbf{Z}_v; \mathbf{Z}_u, \boldsymbol{\theta}_T), \mathbf{y}_v\Big). \tag{5}$$

Note that the teacher's predictive variance $\sigma^2_{\boldsymbol{\theta}_T}(\mathbf{z}_{u_i})$ for the $i$-th pseudo label is added to the $i$-th diagonal entry of the covariance matrix $\mathbf{K}$ in Eq. (3). When $\sigma^2_{\boldsymbol{\theta}_T}(\mathbf{z}_{u_i})$ is significantly greater than that of other pseudo labels, the corresponding diagonal element $\mathbf{K}_{ii}$ can be much larger than the other diagonal elements. As a result, per Eq. (4) the contributions of the $i$-th pseudo label $\mu_{\boldsymbol{\theta}_T}(\mathbf{z}_{u_i})$ to the posterior mean predictions of the validation labels are considerably reduced, indicating that our student model's predictions are less dependent on pseudo labels with large teacher uncertainty.

## 4.4 BI-LEVEL OPTIMIZATION OF THE TEACHER-STUDENT MODEL

Due to the dependence of the student on the teacher via the teacher's pseudo labels, we jointly optimize both by solving a bi-level optimization problem:

$$\begin{aligned} \min_{\boldsymbol{\theta}_T} &\left\{ \mathcal{L}_l(\mathbf{Z}_l, \mathbf{y}_l; \boldsymbol{\theta}_T) + \lambda \mathcal{L}_f\big(\mathbf{Z}_v, \mathbf{y}_v; \boldsymbol{\theta}_S^*(\boldsymbol{\theta}_T), \boldsymbol{\theta}_T, \mathbf{Z}_u\big) \right\}, \\ s.t. \quad &\boldsymbol{\theta}_S^*(\boldsymbol{\theta}_T) = \operatorname*{argmin}_{\boldsymbol{\theta}_S} \mathcal{L}_u(\mathbf{Z}_u, \boldsymbol{\theta}_T; \boldsymbol{\theta}_S). \end{aligned} \tag{6}$$

On the lower level, the student is optimized by minimizing the NMLL ($\mathcal{L}_u$) of Eq. (2), given the teacher's parameters $\boldsymbol{\theta}_T$. On the upper level, the teacher is optimized based on two losses: the Negative Log-Likelihood (NLL) loss $\mathcal{L}_l$ on the labeled dataset $\mathcal{D}_l$ and the feedback loss $\mathcal{L}_f$ from the student, which is weighted by a hyperparameter $\lambda > 0$. $\mathcal{L}_l$ is defined as [3]:

$$\mathcal{L}_l(\mathbf{Z}_l, \mathbf{y}_l; \boldsymbol{\theta}_T) := \frac{1}{N} \sum_{\{\mathbf{z}_l, y_l\} \subset \{\mathbf{Z}_l, \mathbf{y}_l\}} \frac{1}{2} \left( \ln\big(\sigma^2_{\boldsymbol{\theta}_T}(\mathbf{z}_l)\big) + \frac{\big(y_l - \mu_{\boldsymbol{\theta}_T}(\mathbf{z}_l)\big)^2}{\sigma^2_{\boldsymbol{\theta}_T}(\mathbf{z}_l)} \right) + \text{const.} \tag{7}$$

We adopt a computationally efficient alternating one-step gradient-based approximation method to solve Eq. (6). In every training epoch, we first perform a one-step update of the student using the gradient $\nabla_{\boldsymbol{\theta}_S} \mathcal{L}_u$, and then fix the student and update the teacher for one step using the gradient $\nabla_{\boldsymbol{\theta}_T} (\mathcal{L}_l + \lambda \mathcal{L}_f)$. This approach is summarized in Algorithm 1.

## 5 OPTIMIZED UNLABELED DATA SAMPLING STRATEGIES

Random sampling is often utilized in non-BO settings to determine the locations of unlabeled data. However, this may create several issues, particularly under BO. First, the teacher may generate pseudo labels with poor quality at random locations far away from the training data, which may mislead the student and eventually lead to inferior performance. Moreover, evaluating the performance of the student that is trained with random unlabeled data far away from the global optimum, may not provide relevant feedback for tuning the teacher toward finding the global optimum. To this end, we propose two techniques that offer an optimized sampling distribution for unlabeled data. Our experiment in Section 6.4 demonstrates the inferior performance of uniform sampling, underscoring the necessity of systematically sample strategies for unlabeled data.

### 5.1 EXTREME VALUE THEORY (EVT) BASED UNLABELED DATA SAMPLING

To address the issues resulting from random sampling, we develop a guided generation of unlabeled data based on Extreme value theory (EVT) (Fisher & Tippett, 1928). The key idea is to place unlabeled data in regions of high-quality pseudo labels and at the same time encourage exploration

---

[2]The feedback loss can be an MSE, a negative predictive marginal log-likelihood (Gneiting & Raftery, 2007) or a negative Mahalanobis distance (Bastos & O'hagan, 2009). For numerical stability, we choose the MSE in our work.

[3]To ensure numerical stability, we follow (Lakshminarayanan et al., 2017) and warp the variance with the *softplus* function plus a small value: $\sigma^2_{\boldsymbol{\theta}_T} \leftarrow \ln\big(1 + \exp(\sigma^2_{\boldsymbol{\theta}_T})\big) + 10^{-6}$.

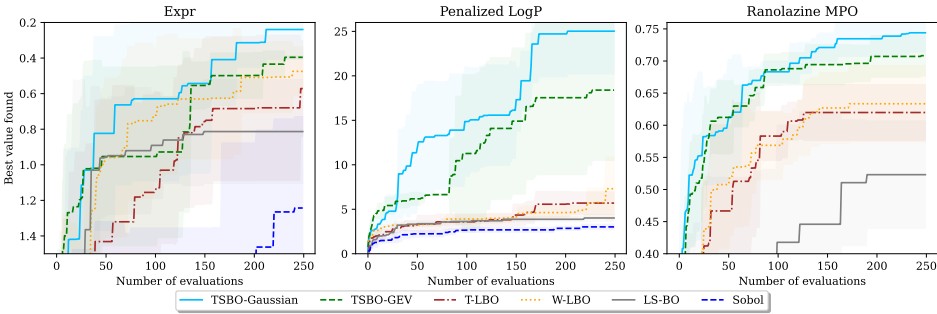

Figure 2: Mean performance and standard deviations of 4 LSO baselines and `TSBO`.

towards the global optimum. To do so, we model the distribution of the part of the labeled data, which are *extreme*, i.e. with the best target values. EVT states that if $\{y_1, \cdots, y_N\}$ are i.i.d. and as $N$ approaches infinity, their maximum $y^*$ follows a generalized extreme value (GEV) distribution (Fisher & Tippett, 1928)

$$p_{y^*}(y^*) = \mathbb{I}_{\{\xi \neq 0\}} (1 + \xi\bar{y})^{-\frac{1}{\xi}} e^{-(1+\xi\bar{y})^{-\frac{1}{\xi}}} + \mathbb{I}_{\{\xi=0\}} e^{-\bar{y}} e^{-e^{-\bar{y}}}, \tag{8}$$

where $\bar{y} := (y^* - a)/b$ defined by 3 learnable parameters of the GEV distribution: a location coefficient $a \in \mathbb{R}$, a scale value $b > 0$, and a distribution shape parameter $\xi \in \mathbb{R}$.

We fit a GEV distribution $p_{y^*}$ with parameters estimated by minimizing the NLL loss of several extreme labels. This GEV distribution captures the distribution of the best-observed target values as seen from the current evaluated data. As such, generating unlabeled data whose predicted labels follow the GEV distribution allows us to start out from the region of the existing extreme labeled data while exploring points with potentially even greater target values due to the random nature of the sampling process. Once the GEV distribution $p_{y^*}$ is fitted, we adopt a Markov-Chain Monte-Carlo (MCMC) method (Hu et al., 2019) to sample from it.

## 5.2 UNLABELED DATA SAMPLING DISTRIBUTION LEARNED FROM STUDENT'S FEEDBACK

While the proposed GEV distribution approach offers a theoretically sound method for generating unlabeled data, its practical effectiveness is constrained by the computationally intensive nature of the MCMC sampling technique (Andrieu et al., 2003).

To circumvent the computational burden associated with MCMC, we endeavor to identify an alternative approach for sampling unlabeled data, denoted as $z_u$, from a distribution $p_{z_u}(\cdot; \theta_u)$ parameterized $\theta_u$. In this pursuit, we turn to the reparametrization trick (Kingma & Welling, 2013) as our preferred sampling strategy. By introducing a random vector $r \in \mathcal{R} \subseteq \mathbb{R}^d$ and a mapping function $g(\cdot; \theta_u) : \mathcal{R} \to \mathcal{Z}$, where $g(r; \theta_u) \sim p_{z_u}$ when $r \sim p_r$, we can efficiently sample unlabeled data $z_u := g(r; \theta_u)$ using $p_r$, a known distribution that can be conveniently sampled from, such as a Gaussian distribution.

Furthermore, we propose an approach to optimize the sampling distribution $p_{z_u}$ and seamlessly integrate it into the teacher-student paradigm. Learning a parameterized sampling distribution by minimizing the feedback loss is a sensible choice. A large feedback loss is indicative of the use of unlabeled data with poor pseudo-label quality, which can potentially mislead the teacher-student model. We optimize $\theta_u$ to minimize the feedback loss $\mathcal{L}_f$:

$$\theta_u^* = \underset{\theta_u}{\operatorname{argmin}} \, \mathbb{E}_{Z_u \sim p_{z_u}} \mathcal{L}_f(Z_v, y_v; \theta_S, \theta_T, Z_u). \tag{9}$$

The gradient for updating $\theta_u$ can be expressed using the reparametrization trick as follows:

$$\nabla_{\theta_u} \mathbb{E}_{Z_u \sim p_{z_u}} \mathcal{L}_f(Z_v, y_v; \theta_S, \theta_T, Z_u) = \nabla_{\theta_u} \mathbb{E}_{R \sim p_r} \mathcal{L}_f(Z_v, y_v; \theta_T, \theta_S, g(R; \theta_u)), \tag{10}$$

where $R \in \mathbb{R}^{M \times d}$ is a batch of $M$ samples $\{r_i\}_{i=1}^M$. We incorporate the update of $\theta_u$ to the alternating one-step scheme for $\theta_S$ and $\theta_T$ described in Section 4.4, as detailed in Appendix A.

Table 1: Mean and standard deviation of the best value after 250 new queries

| Method | Expression ($\downarrow$) | Penalized LogP ($\uparrow$) | Ranolazine MPO ($\uparrow$) |
|---|---|---|---|
| Sobol | 1.261±0.689 | 3.019±0.296 | 0.260±0.046 |
| LS-BO | 0.579±0.356 | 4.019±0.366 | 0.523±0.084 |
| W-LBO | 0.475±0.137 | 7.306±3.551 | 0.633±0.059 |
| T-LBO | 0.572±0.268 | 5.695±1.254 | 0.620±0.043 |
| TSBO-GEV | 0.396±0.07 | 18.40±7.890 | 0.708±0.032 |
| TSBO-Gaussian | **0.24±0.168** | **25.02±4.794** | **0.744±0.030** |

Table 2: A broader comparison on the Chemical Design Task to maximize the Penalized LogP

| Method | $n_{\text{Init}}$ | $n_{\text{Query}}$ | Penalized LogP ($\uparrow$) | Best value ($\uparrow$) |
|---|---|---|---|---|
| T-LBO | 200 | 250 | 5.695±1.254 | 7.53 |
| | | 500 | 10.824±4.688 | 16.45 |
| | 2500 | 5,250 | N/A | 38.57 |
| | 250,000 | 500 | 26.11 | 29.06 |
| TSBO-Gaussian | 200 | 250 | 25.02±4.794 | 31.67 |
| | | 500 | 28.04±3.731 | 32.92 |

## 6 EXPERIMENTAL RESULTS

We aim to empirically demonstrate 1) the overall sample efficiency of `TSBO`, 2) the enhanced generalization capabilities of the data query GP model when incorporating pseudo labels, and 3) the effectiveness of each proposed technique.

### 6.1 EXPERIMENTAL SETTINGS

We employ `TSBO` in 3 challenging high-dimensional global optimization benchmarks, based on two datasets: 40K single-variable arithmetic expressions (Kusner et al., 2017) for an arithmetic expression reconstruction task, and 250K molecules (ZINC250K) (Sterling & Irwin, 2015) for two chemical design tasks. For the chemical design tasks, two objective molecule profiles are considered, respectively: the penalized water-octanol partition coefficient (Penalized LogP) (Gómez-Bombarelli et al., 2018), and the Ranolazine MultiProperty Objective (Ranolazine MPO) (Brown et al., 2019). Detailed descriptions of these three tasks are shown in Appendix B.2.

**Baseline Methods** To assess the efficacy of `TSBO`, we conduct a comprehensive comparative analysis against three VAE-based latent space optimization baselines: LS-BO (Gómez-Bombarelli et al., 2018), W-LBO (Tripp et al., 2020), and T-LBO (Grosnit et al., 2021). Additionally, we include the random search algorithm Sobol (Owen, 2003) for reference. LS-BO performs BO in the latent space with a fixed pre-trained VAE; W-LBO periodically fine-tunes the VAE with current labeled data; T-LBO introduces deep metric learning to W-LBO by additionally minimizing the triplet loss of the labeled data, and is one of the present best-performing methods. We follow the setups described in their original papers (Tripp et al., 2020; Grosnit et al., 2021; Gómez-Bombarelli et al., 2018).

**`TSBO`'s Details** `TSBO` is constructed over the baseline T-LBO, whose methodology is hose methodology is concisely outlined in Appendix B.4. The only difference between `TSBO` and T-LBO is the data query GP: in our case, it is fitted on labeled data and unlabeled data with predicted pseudo labels. We denote `TSBO` with the optimized Gaussian distribution of unlabeled data sampling by TSBO-Gaussian, and `TSBO` with the GEV distribution for sampling unlabeled data by TSBO-GEV. More details about `TSBO` configurations are listed in Appendix B.3.

**Experimental Setup** Our initial labeled data is limited to 100 for the arithmetic expression reconstruction task, and 200 for the two chemical design tasks, respectively. Different from the initialization in (Tripp et al., 2020; Grosnit et al., 2021) which utilizes the full datasets for initialization, we only allow access to no more than 1‰ of the labeled data in the chemical design tasks and 1% of the arithmetic expression task, creating test cases under a tight total labeled data budget. To reduce the performance fluctuations induced by random initialization, we repeat each experiment over 5 random seeds and report the mean performance and its standard deviation.

Table 3: The NMLL loss on testing data

| Data query model | Expression | Penalized LogP | Ranolazine MPO |
|---|---|---|---|
| GP w/o pseudo labels | 1.055 | 0.881 | -1.504 |
| GP with pseudo labels | **0.650** | **0.863** | **-2.019** |

Table 4: The ablation test of `TSBO` on the Chemical Design Task with 250 new queries

| Method | PL predictor | Student | $\mathbf{Z}_u$ Sampling | Penalized LogP ($\uparrow$) |
|---|---|---|---|---|
| T-LBO | - | - | - | 5.695±1.25 |
| T-LBO with PL | MLP | - | Gaussian | 9.917±6.251 |
| | Prob-MLP | - | Gaussian | 17.557±6.998 |
| | Oracle | - | Gaussian | 26 @ 100 queries |
| `TSBO` | MLP | GP | Optimized Gaussian | 12.568±7.965 |
| | Prob-MLP | GP | Optimized Gaussian | 21.115±6.382 |
| | Prob-MLP | UA-GP | Uniform | 4.881±1.416 |
| | Prob-MLP | UA-GP | Gaussian | 23.464±9.535 |
| | Prob-MLP | UA-GP | Optimized Gaussian | 25.02±4.79 |

## 6.2 Efficacy of the Proposed `TSBO`

As shown in Fig. 2, both TSBO-GEV and TSBO-Gaussian consistently outperform T-LBO and other baselines across all evaluated problems within 250 data evaluations. Notably, TSBO-Gaussian demonstrates the ability to discover high target values using a small amount of queried data at the beginning of the BO process. Table 1 provides a summary of the mean performances and standard variations, demonstrating `TSBO`'s superior performance.

Furthermore, our experiments underscore the sample efficiency of `TSBO`. Table 2 shows the best penalized LogP score acquired by T-LBO and TSBO-Gaussian with different numbers of initial data and new queries. Remarkably, even when initiating the process with less than 1‰ of the available samples from the ZINC 250K dataset, TSBO-Gaussian surpasses T-LBO with the utilization of the full dataset after 500 new queries, indicating `TSBO`'s significant sample efficiency.

## 6.3 Mechanism of `TSBO`: Improved Generalization of Data Query GP Model

We analyze how pseudo labels benefit the data query model. After 250 new queries in all 3 optimization tasks, we sample 100 test examples from the standard Gaussian distribution in the latent space. Then, we compare the NMLL loss of posterior predictions for the testing data between a GP fitted exclusively on labeled data, and another GP fitted on both labeled data and unlabeled data with pseudo labels predicted by TSBO-Gaussian. As shown in Table 3, pseudo labels reduce the GP error on testing data, indicating `TSBO` improves the generalization ability of the data query model.

## 6.4 Benefits of Pseudo-Label Prediction and Unlabeled Data Sampling

We conducted an ablation study to assess the efficacy of the proposed techniques within `TSBO`, namely: (1) the benefit of pseudo-label prediction, 2) the improvement of introducing a teacher-student model, 3) the usefulness of the proposed uncertainty-aware teacher-student model, and 4) the necessity of unlabeled data sampling. In addition, Appendix C demonstrates the robustness of our approach to the selection of the feedback weight $\lambda$.

The results of the ablation study on the proposed techniques within `TSBO` are presented in Table 4. We denote PL as pseudo labels, Prob-MLP as an MLP with both mean and variance output (same architecture to the teacher in Section 6.1), Oracle as the true evaluation function, and UA-GP as the proposed pseudo-label dependent uncertainty-aware GP in Section 4.2. For all variants of `TSBO`, we set $\lambda$ to 0.1.

**The Benefit of Pseudo-Label Prediction** For the baseline T-LBO, pseudo-label prediction improves the best function evaluations after 250 queries, demonstrating the effectiveness of pseudo-label prediction. Furthermore, the oracle predictor achieves superior performance, demonstrating the significance of introducing better training strategies for pseudo-label predictors.

**The Improvement of Introducing a Teacher-Student Model** Comparing the method T-LBO with PL to `TSBO`, we observe that incorporating a student model, irrespective of whether the predictor is an MLP or a Prob-MLP, consistently leads to improved performance. This observation indicates that the feedback loss mechanism aids the teacher in generating more accurate pseudo labels.

**The Importance of Uncertainty-Awareness in the Teacher-Student Model** In TSBO-Gaussian whose teacher is Prob-MLP, the introduction of a UA-GP student rather than a GP results in a noteworthy 18.5% increase in the mean of the Penalized LogP score while simultaneously reducing the standard deviation by 25.0%, highlighting the efficacy of the proposed teacher-student model.

**The Necessity of Unlabeled Data Sampling** In `TSBO`, uniform sampling of unlabeled data yields inferior performance compared to the baseline T-LBO, underscoring the significance of the sampling strategy. Moreover, for the Gaussian distribution, the proposed hierarchical optimization method of the teacher-student and the distribution achieves superior results in terms of both mean and variance.

## 7 RELATED WORKS

**Latent Space BO** BO suffers from its high computational complexity, particularly for high-dimensional problems. To sidestep the challenge, the common practice is to perform BO in a reduced latent space. Unsupervised latent space approaches learn a latent representation from abundant unlabeled data with a linear random matrix (Wang et al., 2016), a nonlinear embedding(Moriconi et al., 2020), or a VAE (Kusner et al., 2017), however, without leveraging the rich information of labeled data. Among supervised latent space BO approaches that operate on labeled data, (Tyagi & Cevher, 2014) samples an embedding matrix using low-rank matrix recovery while MGPC-BO (Moriconi et al., 2020) optimizes a nonlinear dimension reduction model, consisting of a perceptron (Rosenblatt, 1958) as an encoder and a multi-output GP (Alvarez & Lawrence, 2008) as a decoder. Without employing unlabeled data, the performance of supervised latent space BO techniques is severely limited by the lack of labeled data in high-dimensional spaces.

**Semi-supervised Learning for Latent Space BO** Semi-supervised approaches address the limitations of the aforementioned techniques by exploring both labeled and unlabeled data. SILBO (Chen et al., 2020) makes use of the spatial information of unlabeled data to learn a linear embedding matrix using semi-supervised discriminant analysis (Cai et al., 2007) and slice inverse regression (Li, 1991). Linear embeddings, however, offer limited dimension reduction performance. More powerful nonlinear projection methods such as W-LBO (Tripp et al., 2020) train a variational autoencoder (VAE) with unlabeled data while updating the VAE when additional labeled data become available. T-LBO (Grosnit et al., 2021), one of the recent competitive approaches, improves W-LBO by introducing deep metric learning (Xing et al., 2002) to pull the labeled data with similar target values together in the latent space. The use of unlabeled data is limited to dimension reduction in W-LBO (Tripp et al., 2020) and T-LBO (Grosnit et al., 2021), in which no predicted pseudo labels are explored.

Differently, the proposed semi-supervised `TSBO` aims to integrate unlabeled data with their predicted pseudo labels into the core optimization steps of BO, apart from performing dimension reduction. Exploring predicted pseudo labels allows training better data query GP models with unlabeled data, mitigating the challenges brought by the sparsity of labeled data.

## 8 CONCLUSION

For the first time, we propose `TSBO` introducing a pseudo-label dependent uncertainty-aware teacher-student model for semi-supervised learning in the context of Bayesian optimization (BO) to improve the generalization of the data query model. Critically, we dynamically optimize the unlabeled data sampling from two kinds of probabilistic distributions: a Gaussian distribution obtained by minimizing the feedback loss, and a GEV distribution based on the extreme value theory. Experiments on 3 high-dimensional BO benchmarks show that `TSBO` achieves superior performance in comparison with other competitive latent space BO algorithms under tight labeled data budgets.

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

# A    ALTERNATING ONE-STEP UPDATE RULE

When unlabeled data $\mathbf{Z}_u$ are sampled from the distribution $p_{\mathbf{z}_u}(\cdot|\boldsymbol{\theta_u})$, we adopt the reparameterization trick to optimize $\boldsymbol{\theta_u}$. In the $i$th training iteration of the teacher-student, we update $\boldsymbol{\theta_u}^{i-1}$ with a learning rate $\eta_u$:

- Sample $\mathbf{Z}_u$ with reparameterization trick: $\mathbf{Z}_u = g(\mathbf{R}, \boldsymbol{\theta_u}^{i-1})$ where $\mathbf{R} \sim p_{\mathbf{r}}$;

- Update $\boldsymbol{\theta}_S^i$ and $\boldsymbol{\theta}_T^i$ as introduced in Algorithm 1;

- Fix $\boldsymbol{\theta}_S^i$, and update $\boldsymbol{\theta_u}^i$: $\boldsymbol{\theta_u}^i \leftarrow \boldsymbol{\theta_u}^{i-1} - \eta_u \nabla_{\boldsymbol{\theta_u}^{i-1}} \mathcal{L}_f(\mathbf{Z}_v, \mathbf{y}_v; \boldsymbol{\theta}_T^i, \boldsymbol{\theta}_S^i, \mathbf{Z}_u)$.

# B    EXPERIMENTAL DETAILS

We have made our code openly available[4].

## B.1    DYNAMIC SELECTION OF VALIDATION DATA

It is attempting to use the full set of available labeled data $\mathcal{D}_l$ as $\mathcal{D}_v$ to assess the student, as proposed in (Pham et al., 2021) for image classification proposes. However, it is not always optimal under the setting of BO, whose objective is to find the global optimum using an overall small amount of labeled data. Hence, the assessment of the student, which provides feedback to the teacher, shall be performed in a way to improve the accuracy of the teacher-student model in predicting the global optimum. Since the majority of labeled data $\mathcal{D}_l$ are used in training the teacher, the quality of pseudo labels around $\mathcal{D}_l$ is high. Thus, validating the student using $\mathcal{D}_l$ may lead to a low averaged loss, however, which is not necessarily indicative of the model's capability in predicting the global optimum. Our empirical study shows that the performance of TSBO improves as the validation data are chosen to be the ones with higher target values. This is meaningful in the sense that assessing the student in regions with target values closer to the global optimum provides the best feedback to the teacher for improving its accuracy at places where it is most needed. We adopt a practical way to dynamically choose $\mathcal{D}_v$ at each BO iteration: the subset of $\mathcal{D}_l$ with the $K$ highest label values. For this, we apply a fast sorting algorithm to rank all labeled data.

## B.2    HIGH-DIMENSIONAL OPTIMIZATION TASKS

**Arithmetic Expression Reconstruction Task:** The objective is to discover a single-variable arithmetic expression $\mathbf{x}^* = $ `1 / 3 + v + sin( v * v )`. For an input expression $\mathbf{x}$, the objective function is a distance metric $f(\mathbf{x}) = \max\{-7, -\log(1 + \text{MSE}(\mathbf{x}(\mathbf{v}) - \mathbf{x}^*(\mathbf{v}))\}$, where $\mathbf{v}$ are 1,000 evenly spaced numbers in $[-10, 10]$. A grammar VAE (Kusner et al., 2017) with a latent space of dimension 25 is adopted. It is pre-trained on a dataset of 40,000 expressions (Kusner et al., 2017).

**Chemical Design Task:** The purpose of this task is to design a molecule with a required molecular property profile. The objective profiles considered are 1) the penalized water-octanol partition coefficient (Penalized LogP) (Gómez-Bombarelli et al., 2018), and 2) the Ranolazine Multiproperty Objective (Ranolazine MPO) (Brown et al., 2019). A Junction-Tree VAE (Jin et al., 2018) with a latent space of dimension 56 and pre-trained on the ZINC250k dataset (Sterling & Irwin, 2015).

For each task, prior to optimization, a VAE is pre-trained using unlabeled data through the maximization of the ELBO (Kingma & Welling, 2013), and all methods employ this pre-trained VAE at the outset of optimization.

## B.3    TSBO'S MODEL ARCHITECTURE AND HYPERPARAMETERS

In TSBO, the teacher model is a multilayer perceptron (MLP) with 5 hidden layers and ReLU activation (Nair & Hinton, 2010). The output dimension is 2. The student model is a standard GP with an RBF kernel.

For the purpose of reproducibility, we provide a comprehensive account of the hyperparameters employed in all our experiments using TSBO. Our approach is based on T-LBO, and thus we adopt

---

[4] https://anonymous.4open.science/r/TSBO-Official-B67E

the default hyperparameters as suggested by (Grosnit et al., 2021). The remaining hyperparameters, specific to `TSBO`, are presented in Table 5.

| | Hyper-parameter | Topology | Expr | Chem |
|---|---|---|---|---|
| | Number of training steps in each BO iteration | 20 | 20 | 20 |
| | Number of warm-up steps | 2,000 | 2,000 | 2,000 |
| | Feedback weight | $10^{-4}$ | $10^{-1}$ | $10^{-1}$ |
| | Number of validation data | 10 | 10 | 30 |
| Common | Number of sampled unlabeled data | 100 | 100 | 300 |
| | Acquisition | EI | EI | EI |
| | Acquisition optimizer | LBFGS | LBFGS | LBFGS |
| | Learning rate | $10^{-4}$ | $10^{-3}$ | $10^{-4}$ |
| Teacher | Batch size of labeled data | 256 | 256 | 32 |
| | Optimizer | Adam | Adam | Adam |
| | Kernel | RBF | RBF | RBF |
| Student | Prior mean | Const. | Const. | Const. |
| | Learning rate | $10^{-2}$ | $10^{-2}$ | $10^{-2}$ |
| | Optimizer | Adam | Adam | Adam |
| Data Query GP | Kernel | RBF | RBF | RBF |
| | Prior mean | Const. | Const. | Const. |

Table 5: Hyper-parameters

### B.4 TRAINING OF VAE IN `TSBO`

Although `TSBO` stands as a general BO framework, it has been seamlessly integrated into T-LBO (Grosnit et al., 2021), a state-of-the-art VAE-based BO method, to facilitate a fair comparison. The training approach for the VAE remains unaltered, aligning with T-LBO's methodology:

• Pretrain: Before the first BO iteration, the VAE is trained on the dataset in an unsupervised way to maximize the ELBO (Kingma & Welling, 2013);

• Fine-tune: After each 50 BO iteration, the VAE is trained on all labeled data for 1 epoch to both maximize the ELBO and minimize the triplet loss which penalizes data having similar labels located far away. The weight of triplet loss is set to 10 in the Expression task and 1 in the Chemical Design task.

The training schemes of all models proposed in `TSBO` and the VAE are decoupled, rendering T-LBO an apt baseline for validating `TSBO`'s sample efficacy.

Table 6: The ablation test of the weight of the feedback loss

| Method | $\lambda$ | Expression ($\downarrow$) | Penalized LogP ($\uparrow$) | Ranolazine MPO ($\uparrow$) |
|---|---|---|---|---|
| T-LBO | - | 0.572±0.268 | 5.695±1.254 | 0.620±0.043 |
| | 0.001 | 0.240±0.168 | 21.106±8.960 | 0.713±0.021 |
| TSBO-Gaussian | 0.01 | 0.433±0.260 | 21.384±1.533 | 0.720±0.040 |
| | 0.1 | 0.450±0.130 | 25.021±4.794 | 0.744±0.030 |
| | 1 | 0.474±0.113 | 21.122±7.494 | 0.712±0.023 |

## C  The Influence the Feedback Weight To `TSBO`

We analyze the influence of the selection of the feedback weight $\lambda$. Our experiments demonstrate that in a large range of $\lambda$, `TSBO` consistently outperforms the baseline T-LBO, underscoring the robustness of our approach to this hyper-parameter.

As shown in Table 6, while the selection of $\lambda$ in $\{0.001, 0.01, 0.1, 1\}$ has an impact on `TSBO`'s performance, for each considered $\lambda$, TSBO-Gaussian consistently outperforms T-LBO, indicating that our success is not contributed to a deliberate $\lambda$ selection.

## D  Additional Ablation Test of Validation Data Selection

In order to verify the effectiveness of the proposed dynamic selection of validation data $\mathcal{D}_v$ in `TSBO`, where $\mathcal{D}_v$ is the subset of $\mathcal{D}_l$ with the $K$ highest label values, we conduct an ablation study to compare it (TSBO-Gaussian) with two non-optimized $\mathcal{D}_v$ selection strategies: random $K$ examples uniformly sampled from $\mathcal{D}_l$ (TSBO-Gaussian-ValRand), and the current labeled dataset $\mathcal{D}_l$ (TSBO-Gaussian-ValAll). These three variants of `TSBO` are measured on the Chemical Design task, and we report their average best values of 5 runs starting with 200 initial labeled data and a new data query budget of 250. $K$ is set to 30.

As shown in Table 7, despite each variant of `TSBO` outperforms the baseline T-LBO, both TSBO-Gaussian-ValRand and TSBO-Gaussian-ValAll are less competitive than TSBO-Gaussian in finding the global maximum. This result meets our expectations: the student's feedback on those examples with the $K$ highest label values is more beneficial for the training of the teacher-student model, and eventually better facilitating the search for the maximum.

Table 7: Comparison of validation data selection of `TSBO` on the Chemical Design task

| Method | Validation Data Selection | Best Value ($\uparrow$) |
|---|---|---|
| T-LBO | - | 5.70 |
| TSBO-Gaussian-ValRand | Random 30 | 21.60 |
| TSBO-Gaussian-ValAll | All labeled data | 22.65 |
| TSBO-Gaussian | Top 30 | **25.02** |

## E  Robustness of `TSBO` to Noisy Labels

In this section, we demonstrate the robustness of `TSBO` in noisy environments. We compare `TSBO` with the other baselines on the Chemical Design task, where all labels are subject to additional i.i.d. zero-mean Gaussian noises, with a standard deviation (std) of 0.1. All methods start with 200 initial labeled data and query 250 new examples. We report the mean and the standard deviation of the best values found by each method among 5 runs in Table 8.

Table 8: Labels with white Gaussian noises on the Chemical Design Task

| method | Best Penalized LogP ($\uparrow$) | |
|---|---|---|
| | Noise variance=0 | Noise variance=0.1 |
| Sobol | 3.019±0.296 | 3.06±0.38 |
| LS-BO | 4.019±0.366 | 4.41±0.68 |
| W-LBO | 7.306±3.551 | 4.44±0.27 |
| T-LBO | 5.695±1.254 | 4.22±0.68 |
| TSBO-GEV | 18.40±7.89 | 20.73±6.24 |
| TSBO-Gaussian | **25.02±4.794** | **25.97±4.82** |

While nearly all of the baselines, especially T-LBO, exhibit decreases in the noisy scenario compared to the results obtained without observation noise, `TSBO` shows no performance deterioration. This phenomenon demonstrates the noise-resistant capability of the proposed uncertainty-aware teacher-student model.

## F    BROADER IMPACT

The proposed `TSBO` has the potential for significant positive impacts in various domains. By effectively finding the optimum compared with baselines on multiple datasets, `TSBO` offers a promising solution to enhance the efficiency and efficacy of optimization processes given limited labeled data and evaluation budgets. For instance, in engineering and manufacturing, `TSBO` can facilitate outlier detection, failure analysis, and the design of more efficient processes, leading to increased productivity and reduced resource consumption. By enabling faster and more accurate optimization, `TSBO` can ultimately benefit society as a whole.

Even though `TSBO` holds great promise, it is important to acknowledge and mitigate potential negative impacts. One concern is the overreliance on automated optimization algorithms, which could lead to a decreased emphasis on human intuition and creativity. `TSBO` should be used as a supportive tool that enhances human decision-making rather than replacing it entirely. Additionally, there is a risk of bias in the optimization process if the training data used for the teacher model contain inherent biases. Careful attention must be given to the training data to ensure fair and unbiased optimization results.

In conclusion, `TSBO` offers significant potential for broad impact in optimization tasks. By improving the efficiency and efficacy of optimization processes, `TSBO` can accelerate the discovery of optimal solutions, benefiting various industries and ultimately improving the well-being of individuals and society at large. However, it is important to consider and mitigate potential negative impacts, such as overreliance on automation and the risk of bias, to ensure that `TSBO` is used responsibly and ethically. With proper safeguards and considerations, `TSBO` can be a valuable tool that enhances human expertise and drives advancements in optimization across diverse domains.

