# OpenReview forum: "High-dimensional Bayesian Optimization via Semi-supervised Learning with Optimized Unlabeled Data Sampling"
_ICLR.cc/2024/Conference — Submitted to ICLR 2024_

### Official Review · Reviewer_r1KM · 2023-10-30

**Soundness:** 2 fair
**Presentation:** 2 fair
**Contribution:** 3 good
**Rating:** 5
**Confidence:** 4

**Summary:**

The paper proposes TSBO, a Bayesian optimization algorithm that employs semi-supervised learning through a teacher-student model which generates optimized synthetic data, which is incorporated into  a conventional surrogate model. The aim is to improve surrogate model generalization, and thereby enhance sample efficiency. The semi-supervised learning process involves a bi-level optimization scheme, which iteratively optimizes the mariginal log likelihood of synthetic data of the student and a weighted sum of. Results show improved performance on three chemical design tasks.

**Strengths:**

__Novel idea:__ Self-supervised learning has not thoroughly been explored in a BO context, and so the proposed idea explores a novel niche.

__Addressing model accuracy:__ Addressing BO efficiency from the standpoint of producing a more accurate surrogate is a good approach - one that I believe deserves attention.

__Multiple relevant ablations:__ With a large number of components to the algorithm, substantial resources have been dedicated towards ablations.

__Figure 1:__ This effectively communicates the proposed method in a clear and pedagogical manner.

**Weaknesses:**

While the proposed approach is interesting, I believe that there is a substantial amount of complexity that is currently unjustified. Moreover, little is offered in terms of intuition as to why the involved components are jointly able to produce a more accurate surrogate model. I believe the paper has potential, but these outstanding issues would have to be addressed in a satisfactory manner for the paper to be accepted. Specifically,

(To avoid confusion, I will denote the BO surrogate as the BO GP and the student as the SGP)

### __1. The complexity of the method:__
In addition to the Vanilla GP, there are three learnable components.
- The teacher, trained on two real and synthetic data based on its own classification ability and the student's performance
- The student, trained on synthetic data and validated on real data (synthetic and real)
- The synthetic data generation process, which is optimized to minimize the feedback loss of the first two

This nested scheme makes it difficult to discern whether synthetic data are sensible, are classified correctly and what drives performance - essentially why this process makes sense. As such, plots demonstrating the fit of the student model, the synthetically generated data or anything else which boosts intuition is paramount. As of now, the method is not sufficiently digestible.



### __2. The concept of adding synthetic data:__
 In BO, The (GP) surrogate models the underlying objective in a principled manner which yields calibrated uncertainty estimates, assuming that it is properly trained. Since the GP in itself is principles, I am not convinced that adding synthetic data is an intrinsically good idea. The proposed approach suggests that the GP is either un-calibrated or does not sufficiently extrapolate the observed data. While this seems possible, there is no insight into which possible fault (of the GP) is being addressed, nor how the added synthetic data accomplishes it. Is it
1. The teacher MLP that aggressively (and accurately) extrapolates in synthetic data generation?
2. That the vanilla GP is simply under-confident, so that adding synthetic data acts to reduce the uncertainty by adding more data?
3. Some other reason/combination of the two

Note that I am specifically asking for intuition as to how the _modeling_ (and not the BO results) can improve by adding synthetic data.

### __3. The accuracy and role of the teacher:__
If the teacher is able to generate accurate data (synthetic or real), why not use it as a surrogate instead of the conventional GP? Comments like

#### _"Evaluating the performance of the student that is trained with random unlabeled data far away from the global optimum, may not provide relevant feedback for tuning the teacher toward finding the global optimum."_

suggest that it is the teacher's (and not the BO loop's) task to find the global optimum. If the teacher adds data that it believes is good to the BO surrogate, the teacher is indirectly conducting the optimization.

### __4. The role of the student:__
If the student is trained only on synthetic data, validated on a (promising) subset of the true data, and its loss is coupled with the teacher's. As such, the student's only role appears to be as a regularizer (i.e. to incorporate GP-like smoothness into the MLP) while producing approximately the same predictions on labeled data. Can the authors shed light on the role on whether this is true, and if so, what difference the student offers as opposed to conventional regularization.

### __5. Few tasks in results:__
Three benchmarks constitutes a fairly sparse evaluation. Does the proposed method make sense on conventional test functions, and can a couple of those be added?


### __6. Missing references:__
The key idea of 5.1 is _very_ similar to MES (Wang and Jegelka 2017), and so this is a must-cite. They also use a GEV (specifically a Gumbel) to fit $p(y^*)$.

__Minor:__
- App. B1: proposes -->purposes, remove ", however"
- Sec 5.0 of systematically sample --> of systematically sampling
- Legend fontsize is tiny, Fig.1 fontsize is too small as well

**Questions:**

- Is the synthetically generated data added as with any other data point, or are they added with additional noise?
- Is the synthetically generated data ever replaced or substituted?
- How (qualitatively) does the SGP differ from the BO GP in its prediction?
- Does the parametrized sampling distribution discriminate regions that are believed to be good (like the GEV)?

---

> ### Author Response · Authors · 2023-11-23
> **Response to Reviewer r1KM, Part 1**
>
> We are grateful to the reviewer for their insightful feedback and thorough analysis of TSBO. This has prompted a deeper examination of the roles and interactions of each module within our system. In response to the reviewer's concerns, we present the following points:
>
> ## Response to Weakness
>
> **The concept of adding synthetic data:**
>
> We concur with the reviewer’s suggestion on viewing the proposed teacher-student model, which includes the student’s feedback to the teacher, as a mechanism for providing proper regularization to the teacher. This is a good way of looking at TS-BO. On the other hand, we want to emphasize that the regularization of the teacher is tailored specifically for the sequential optimization in BO. As such, we term it as “selective regularization” to distinguish it from other more commonly used regularizes.
> We elaborate on this “selective regularization” perspective further in the response to Point 4 below.
>
> In response to the current question, we believe that the synthetic data offers advantages in  both areas highlighted by the reviewer:
>
> (a) The "selective regularization" improves the teacher's extrapolation power on synthetic data, more specifically, on the potentially high-value data points. This enhancement is a result of the combined efforts of the teacher, the unlabeled data sampler, and the student model. The sampler generates synthetic data which are potentially of high values. The student model, trained on the synthetic data, provides feedback to the teacher, effectively regularizing the teacher's predictions on these data points.
>
> (b) Vanilla Gaussian Processes (GPs) often exhibit overly conservative behavior when dealing with scarce data, leading to poor extrapolation capabilities [1, 2]. In TSBO, the selective regularization mechanism empowers the teacher to produce more precise synthetic data in regions potentially with high target value. This process increases the accuracy of the predicted label of the synthetic data. As a result, leveraging such synthetic data as additional data to train the data query GP can help reduce its uncertainty in regions that attain a high target value, which well aligns with the goal of Bayesian optimization.

---

> ### Author Response · Authors · 2023-11-23
> **Response to Reviewer r1KM, Part 2**
>
> **The accuracy and role of the teacher:**
>
> This point raised by the reviewer brings up two important issues with respect to the role of teacher which we discussed as below:
>
> (a) Direct utilization of the Teacher Model as the BO Surrogate:
>
> We've conducted an additional experiment to address whether the teacher model can be directly utilized as the surrogate in TSBO. We evaluate three different methods on Expression and Chemical Design. The first method is T-LBO, as the baseline. The second model is the proposed TSBO in the main paper and the third model is a variant of the proposed TSBO where the BO surrogate is the teacher model.
>
> For all model, the hyper-parameter settings are the same as in the paper. The best evaluations of Expr and Penalized LogP after 250 queries among 5 seeds are reported in the table below.
>
> We've also added this experiment with detailed setting in the Appendix in our revised version for further reference.
>
> |Method| Sorrogate|  Expr ($\downarrow$ ) | Penalized LogP ($\uparrow$ ) |
> | - | - | - | -|
> |T-LBO (baseline)          | GP|  0.572±0.268 | 5.695±1.254 |
> |TSBO  | GP| **0.240±0.168** | **25.02±4.794** |
> |TSBO  | Teacher| 0.432±0.235 | 23.574±3.017 |
>
> As shown in the table, even though the teacher are directly adopted as a data query model, TSBO still outperforms T-LBO.
>
> This suggests that the quality of the teacher is better than that of the GP surrogate in more standard BO methods that do not incorporate a teacher-student model.  We could use this teacher to replace the surrogate GP in these more standard BO methods to get better results.
>
> Meanwhile, most of existing BO methods prefer using GPs as the surrogate model due to their principled approach to calibrated uncertainty estimation. Though our teacher model could be more accurate than the standard GP surrogates, its uncertainty calibration is not as robust as that of GPs. Therefore, in our main paper, we proposed to supplement the teacher model with an additional data query GP. This GP incorporates both the synthetic data generated by the teacher and the labeled data. Empirically, this hybrid approach combines the best of the two worlds; it not only retains the improved accuracy of the teacher model but also enhances uncertainty estimation. The above experiment validates that this hybrid approach achieves the best final BO results.
>
> (b) Intuition Behind the Teacher Model's Role:
>
> We concur with the reviewer’s suggestion on viewing the proposed teacher-student model, which includes the student’s feedback to the teacher, as a mechanism for providing proper regularization to the teacher. The teacher’s role can be understood in this "selective regularization" perspective, which will be elaborated in our response to the next question on “The role of the student”.
>
> In brief, we posit that the superiority of the teacher model over traditional GPs as a BO surrogate stems from its enhanced ability to predict the labels of the data in regions of high target value. This is achieved by our BO-specific "selective regularization" of the teacher, implemented by combining the student model with an unlabeled data sampling strategy.

---

> ### Author Response · Authors · 2023-11-23
> **Response to Reviewer r1KM, Part 3**
>
> **The role of the student:**
>
> In agreement with the reviewer's perspective, the proposed teacher-student model, which includes the student’s feedback to the teacher, can be viewed as a mechanism for providing proper regularization to the teacher. On the other hand, we want to emphasize that the regularization of the teacher is tailored specifically for the sequential optimization in BO. As such, we term it as “selective regularization” to distinguish it from other more commonly used regularizes.
>
> The BO context awareness in this “selective regularization” is implemented by incorporating the student’s feedback and an unlabeled data sampling procedure that is optimized for BO.
>
> To achieve the goal of BO, it is desirable to train the teacher model so that it can better predict data points that have high target value. We may achieve this objective by imposing regularization focusing on points of interest in the BO process, rather than indiscriminately across the entire optimization landscape. We achieve this through our unlabeled data sampling technique, which proposes potentially optimal points in the current optimization step. The student model, trained specifically on these points, provides selective feedback to the teacher, effectively fine-tuning it. This process can be thought of as a self-evaluation by the teacher-student model on potential high-value points before performing actual data querying.
>
> **Few tasks in results:**
>
> We evaluate TSBO on an additional challenging high-dimensional global optimization task: Topology Shape Fitting Task, first considered in T-LBO [3]. The goal is to find a pre-defined target binary topology image $\boldsymbol{\mathrm{x}}^* \in \{0, 1\}^{1\times 40\times40}$. The objective function is the cosine similarity between the target $\boldsymbol{\mathrm{x}}^*$ and an input image $\boldsymbol{\mathrm{x}}$: $f(\boldsymbol{\mathrm{x}})=\boldsymbol{\mathrm{x}}^T\boldsymbol{\mathrm{x}}^*/(\lVert \boldsymbol{\mathrm{x}}\rVert_2 \lVert \boldsymbol{\mathrm{x}}^*\rVert_2)$. The VAE used in this task is a convolutional neural network [3] with a latent space of dimension 20. It is pre-trained on 10,000 unlabeled topology pictures in the dataset from [4].
>
>
> |Method | Topology ($\uparrow$ ) |
> | -  | -|
> |Sobol  |  0.781±0.007 |
> |LS-BO  |  0.834±0.021 |
> |W-LBO  |  0.842±0.012 |
> |T-LBO  |  0.840±0.032 |
> |TSBO | **0.862±0.022** |
>
> We compare TSBO with all baselines considered in the manuscript. Each method starts with 100 labeled images, uniformly sampled from the dataset in [4]. We report the best score after 250 new queries, averaged across 3 random seeds. The hyper-parameter settings are the same as in the paper.
>
> As shown in the table above, TSBO significantly outperforms baselines, demonstrating its effectiveness and versatility in global optimization tasks.
>
> **Missing references:**
>
> We will add the mentioned paper in our revised version and provide a review of it.
>
> ## Response to Questions
>
> **1.** The synthetic data are treated as other labeled data, i.e., no additional predicted noises involved in fitting the data query GP.
>
> **2.** Yes, the synthetic data are re-sampled at the beginning of each BO iteration. The computational overhead of sampling is relatively neglectable in light of costly efforts for data evaluations.
>
> **3.** The student GP (SGP) is fitted on synthetic data with pseudo labels. Its accuracy on labeled data tends to be worse than the vanilla BO GP. Thus, the student is only utilized to provide an effective regularization to the teacher via the feedback loss.
>
> **4.** Yes, the data sampled by our parametrized sampling distribution tend to have higher evaluation values than the initial training data or data sampled randomly. For instance, in the final BO iteration for the molecule design task, the highest Penalized LogP value among the synthetic data generated by our sampler is larger than 20, whereas the maximal value of real molecules in the ZINC250 dataset is smaller than 4.
>
> ## Reference
> [1] Kusner, M. J., Paige, B., & Hernández-Lobato, J. M. (2017, July). Grammar variational autoencoder. In International conference on machine learning (pp. 1945-1954). PMLR.
>
> [2] Jin, W., Barzilay, R., & Jaakkola, T. (2018, July). Junction tree variational autoencoder for molecular graph generation. In International conference on machine learning (pp. 2323-2332). PMLR.
>
> [3] Grosnit, A., Tutunov, R., Maraval, A. M., Griffiths, R. R., Cowen-Rivers, A. I., Yang, L., ... & Bou-Ammar, H. (2021). High-dimensional Bayesian optimisation with variational autoencoders and deep metric learning. arXiv preprint arXiv:2106.03609.
>
> [4] Sosnovik, I., & Oseledets, I. (2019). Neural networks for topology optimization. Russian Journal of Numerical Analysis and Mathematical Modelling, 34(4), 215-223.

---

> > ### Comment · Reviewer_r1KM · 2023-12-05
> >
> > Thanks to the authors for their detailed response and additional results.
> >
> > I appreciate the additional details on role of each component of the model. The discussion on selective regularization is a very interesting one, and I would greatly appreciate if it made it into a subsequent version of the paper. This would greatly improve intuition as to why the proposed method works.
> >
> > My perception of the paper (including the rebuttal) is now improved. I have increased my score, but I ultimately believe the paper should be re-worked to incorporate elements outlined here. Specifically, the __qualitative__ impact of each self-supervised component on model quality should be thoroughly discussed, and potentially even visualized for intuition. Coupled with the ablations that are already present, I believe the resulting paper will be very strong.

---

### Official Review · Reviewer_tpuF · 2023-10-31

**Soundness:** 2 fair
**Presentation:** 3 good
**Contribution:** 2 fair
**Rating:** 5
**Confidence:** 4

**Summary:**

The paper considers the problem of developing Bayesian optimization (BO) algorithm for optimizing `high-dimensional` black-box functions. The main proposal is to improve the surrogate modeling component of BO for high-dimensional spaces with a semi-supervised learning approach titled `TSBO` where the key idea is to leverage un-labeled data in the latent space BO framework by learning a pseudo-label dependent uncertainty-aware teacher-student model.

The teacher model is a multilayer perceptron that outputs a (mean, variance) pair which is used by a Gaussian process  based student model. A bilevel optimization problem is defined as the training objective of the student and teacher model. The teacher model parameters are updated with a combination of i. labeled data fitting loss and ii. feedback loss from the optimized student GP model. The student GP model parameters are optimized over the unlabeled dataset. This model is then used to generate the feedback loss for the teacher model based on a labeled validation set. The paper also proposes two different sampling approaches for picking the unlabeled dataset.

Once the teacher-student model is trained, the labeled dataset is combined with teacher-model predicted pseudo-labels for unlabeled inputs to train the main GP surrogate model (referred as data query GP model in the paper) for BO. Experiments are performed on a arithmetic expression task and two chemical design tasks.

**Strengths:**

- The paper considers an important problem with many real-world applications.

- The overall idea of leveraging unlabeled data is nice and a nice broader insight to study as it can be instantiated in multiple different ways.

- Overall BO results on the three domains are promising and the proposed method outperforms existing baseline approaches.

**Weaknesses:**

- I have a broad top-view question. The premise of the paper is very reasonable in suggesting that the bottleneck for existing BO methods is the limited amount of labeled data available. However, labeled data is still required for training all the new components introduced in the paper (teacher model/parametrized sampling distribution). If semi-supervised learning is the key source for getting such improved results, why wouldn't any other semi-supervised learning approach (manifold/laplacian regularization etc) work equally well?

- The ablation described in section 6.3 is critical for showing the performance improvement in surrogate modeling. I think a better way to do it is by evaluating the test performance on original data (for e.g. get a set of molecules from the chemical design and compute their latent embeddings) rather than points sampled from the latent space. In the end, we want the GP surrogate to be better at modeling points which resemble the original input space rather than any general point in the latent space. Please consider running this ablation in this manner.

- There is a large body of work on both high-dimensional BO and latent space BO that is not discussed in the paper. Please see some references below. It is not necessary to compare with them experimentally but they deserve proper discussion in the paper as they are directly relevant to the studied problem. In fact, reference [9] below even considers including semi-supervised learning via an auxiliary network. Reference [6] is one of the key baselines because it re-trains the GP surrogate along with VAE parameters as we get more data.

BO over high dimensional inputs

[1] Eriksson, D., Pearce, M., Gardner, J., Turner, R. D., & Poloczek, M. (2019). Scalable global optimization via local Bayesian optimization. Advances in neural information processing systems, 32.

[2] Eriksson, D., & Jankowiak, M. (2021, December). High-dimensional Bayesian optimization with sparse axis-aligned subspaces. In Uncertainty in Artificial Intelligence (pp. 493-503). PMLR.

[3] Papenmeier, L., Nardi, L., & Poloczek, M. (2022). Increasing the scope as you learn: Adaptive Bayesian optimization in nested subspaces. Advances in Neural Information Processing Systems, 35, 11586-11601.

[4] Letham, B., Calandra, R., Rai, A., & Bakshy, E. (2020). Re-examining linear embeddings for high-dimensional Bayesian optimization. Advances in neural information processing systems, 33, 1546-1558.

[5] Nayebi, A., Munteanu, A., & Poloczek, M. (2019, May). A framework for Bayesian optimization in embedded subspaces. In International Conference on Machine Learning (pp. 4752-4761). PMLR.

Latent space BO

[6] Maus, N., Jones, H., Moore, J., Kusner, M. J., Bradshaw, J., & Gardner, J. (2022). Local latent space bayesian optimization over structured inputs. Advances in Neural Information Processing Systems, 35, 34505-34518.

[7] Deshwal, A., & Doppa, J. (2021). Combining latent space and structured kernels for bayesian optimization over combinatorial spaces. Advances in Neural Information Processing Systems, 34, 8185-8200.

[8] Notin, P., Hernández-Lobato, J. M., & Gal, Y. (2021). Improving black-box optimization in VAE latent space using decoder uncertainty. Advances in Neural Information Processing Systems, 34, 802-814.

[9] Eissman, S., Levy, D., Shu, R., Bartzsch, S., & Ermon, S. (2018, January). Bayesian optimization and attribute adjustment. In Proc. 34th Conference on Uncertainty in Artificial Intelligence.

**Questions:**

Please see weaknesses section above. I am more than happy to increase my score if the questions are answered appropriately.

---

> ### Author Response · Authors · 2023-11-23
> **Response to Reviewer tpuF, Part 1**
>
> We appreciate the reviewers' constructive comments and feedbacks. Based on your suggestions, we provide  detailed responses as follows:
>
> ## Reponses to Weakness
>
> **1.** Manifold Regularization and Laplacian Regularization are predicated on specific data assumptions. For instance, Laplacian Regularization penalizes the norm of the model's gradient when the density of unlabeled data is large.
>
> While these methods can potentially improve the modeling of the data geometry and hence improve the data query process when applied to Bayesian Optimization, the regularizations are applied across the entire data manifold and do not specifically target high-value regions, which are crucial in BO. In contrast, TSBO introduces a more nuanced, context-aware regularization (detailed in the next paragraph), particularly focused on areas with potentially high values. This targeted approach, as we hypothesize, is likely to be more effective within the scope of Bayesian Optimization.
>
> The generation and utilization of unlabeled data within TSBO can be regarded as a form of BO specific "selective regularization" imposed on the teacher model. This enhancement is achieved by the combined efforts of the teacher, the unlabeled data sampler, and the student model: The sampler generates synthetic data which are potentially of high values. The student model, trained on the synthetic data, provides feedback to the teacher, effectively regularizing the teacher's predictions on these data points. This specialized form of regularization enables the teacher model to better extrapolate in regions of potential high values, providing valuable synthetic data that diminishes the uncertainty of the surrogate GP model. Consequently, this leads to improved sample efficiency, as demonstrated in our study.
>
> **2.** As the reviewer suggested, we test the generalization of the data query GP of TSBO for real molecules instead of sampled synthetic data and compare with a GP which is solely trained on  labeled data. The experimental settings are the same as in Section 6.3, expect the testing data. We uniformly sample 100 testing data points from the ZINC250K dataset and report the Negative Marginal Log-likelihood (NMLL, lower the better) of both GPs in the table below.
>
> |Data Query Model   | Penalized LogP |Ranolazine MPO |
> |-| - | - |
> |GP w/o pseudo labels   | 2.196 | -1.186 |
> |GP w pseudo labels  | **2.194** | **-1.197** |
>
> The results show that there is a consistent improved generalization by utilizing pseudo labels for both Penalized LogP and Ranolazine MPO molecule profiles on the real test data points.
>
> **Further discussion:** while GP trained with pseudo labels improves the generalization on the labeled test data, we expect the improvement to be marginal since their corresponding values are very low. As mentioned in the response to weakness 1, the selective regularization targets on improving the teacher's extrapolation performance on potentially high values points, and the surrogate GP model, which utilizes these synthetic data and labeled data, is expected to show improved generalization ability especially on real data with high values.
>
> In this case, an ideal test situation is to evaluate and compare the generalization ability of the TSBO GP and vanilla GP on real data points that of high values. Unfortunately, in the labeled dataset, the maximum value of labeled data is around 4 and the optimized value is 26. In other word, real data that are close to optimal values are not accessible in the provided dataset, making this ideal test difficult to implement in practice.
>
> To further demonstrate the GP surrogate trained with pseudo labels has improved generalization particularly in high value regions, we design another experiment to evaluate the GPs with "synthetic high value" data directly in the latent space, which has not been demonstrated in the previous setting in Section 6.3, where we evaluate the generalization ability over the whole latent space. The setup is detailed as below:
>
> The test data points are 100 synthetic data which are sampled in the latent space, from a small neighborhood of the best synthetic molecule's latent representation after 250 new queries. The sampling distribution is $\mathcal{N} (\mathrm{z}_{opt}, \sigma^2 I_d)$, where $\sigma$ is 0.01. The performance for the two GPs are reported as below:
>
> |Data Query Model   | Penalized LogP |Ranolazine MPO |
> |-| - | - |
> |GP w/o pseudo labels   | 4.228 | -2.086 |
> |GP w pseudo labels   | **1.388** | **-2.391** |
>
> As demonstrated in the preceding table, the introduction of pseudo labels markedly reduces the GP's NMLL loss on testing data. This reduction highlights enhanced generalization, particularly around the optimum points.

---

> ### Author Response · Authors · 2023-11-23
> **Response to Reviewer tpuF, Part 2**
>
> **3.** We appreciate you for mentioning recent reference highly related to TSBO. All of them will be properly introduced in Section "Related Work" of an updated manuscript. Moreover, recent released Latent Space BO work will be discussed as well, such as [1, 2, 3].
>
> Herein, we discuss the two VAE-based Latent Space BO approach that you specifically concerned: BO with attribute adjustment [4] ([9] in your reference order) and LOLBO [5] ([6] in your reference order). In convenience, we denote the first method as AABO.
>
> At a high-level, AABO and LOLBO introduce new loss functions to the VAE in the context of BO. AABO utilizes a target predictor in the latent space, and trains the VAE to simultaneously maximize the standard unlabeled ELBO objective and minimize the supervised MSE loss of the predictor. LOLBO, on the other hand, optimizes its VAE to maximize the standard ELBO and the supervised NMLL loss of the data query GP.
>
> Both approaches aim to leverage semi-supervised learning to train a better VAE, and have achieved promising experimental results. However, TSBO focuses on systematically integrating unlabeled data with their predicted pseudo labels into the data query GP. We believe the proposed techniques in TSBO are complementary to the existing VAE-based Latent Space BO work. Furthermore, [6] and [9] can be integrated into the TSBO framework with ease.
>
> ## Reference
>
> [1] Maus, N., Lin, Z. J., Balandat, M., & Bakshy, E. (2023). Joint Composite Latent Space Bayesian Optimization. arXiv preprint arXiv:2311.02213.
>
>
> [2] Stanton, S., Maddox, W., Gruver, N., Maffettone, P., Delaney, E., Greenside, P., & Wilson, A. G. (2022, June). Accelerating bayesian optimization for biological sequence design with denoising autoencoders. In International Conference on Machine Learning (pp. 20459-20478). PMLR.
>
> [3] Maus, N., Wu, K., Eriksson, D., & Gardner, J. (2022). Discovering Many Diverse Solutions with Bayesian Optimization. arXiv preprint arXiv:2210.10953.

---

### Official Review · Reviewer_QKF7 · 2023-10-31

**Soundness:** 3 good
**Presentation:** 3 good
**Contribution:** 3 good
**Rating:** 6
**Confidence:** 2

**Summary:**

The paper introduces a novel Bayesian Optimization (BO) procedure aimed at improving performances under a high dimensional data and low example setting.
The procedure circumvent the high number of example typically needed in latent space BO by using a "Teacher-Student" update loop to generate pseudo-labels during training. This loop requires sampling unlabeled data which is cheaper.
They empirically demonstrate the efficiency of their procedure and how their proposed optimization of each step of the procedure are key to performance improvement.
The four qualitative improvements demonstrated empirically are the following:
- Pseudo-Label prediction improves performance even for baseline models.
- The teacher-student method is efficient at optimizing the parameters using pseudo labels
- The Uncertainty awareness part of the teacher-student method consistently ensures better performances
- Optimized unlabeled sampling improves performance even for baseline models

**Strengths:**

The method proposed in the article is robust to parameter changes and significantly improves past performances of other models such as .-LBO methods.
The methods trained with limited data even outperforms baseline models with unlimited access to data.
Thus the experiments tend to show that this method paves the way for novel applications where labelled and even unlabelled data is scarce.

**Weaknesses:**

The articles doesn't present the performances of the method with access to larger parts of the dataset. This is coupled with a lack of theoretical analysis of the convergence of the algorithm weakens the presentation of the asymptotic behavior of the method.

**Questions:**

Is there a class of functions for which the method converges towards the maximum?
Can we expect any rate of convergence?

---

> ### Author Response · Authors · 2023-11-23
> **Response to Reviewer QKF7**
>
> We extend our gratitude to the reviewer for the useful concerns raised. In response, we would like to address the following points:
>
> ## Response to Weakness
> We evaluate TSBO starting with more intital data on the Chemical Design task to maximize the Penalized LogP objective. The experimental setting are the same as the paper, except that $n_{init}$ is 400.
>
>
> |Data Query Model|  $n_{init}$ |  $n_{query}$=100 |  $n_{query}$=250|
> |-| - | - | -|
> |T-LBO|200|-|5.695±1.254|
> |T-LBO|250K|-|≤26.11 (at $n_{query}$=500)|
> |TSBO   |200|14.865±6.134| 25.020±4.794 |
> |TSBO  | 400|21.213±12.826| **26.246±2.731** |
>
> The results indicate that more initial labeled data significantly boost TSBO in the beginning 100 evaluations, and improve the final performance and robustness of TSBO. Moreover, TSBO starting with 400 initial data even outperforms T-LBO beginning with the full ZINC250K dataset, demonstrating the superior data efficiency of our method.
>
> ## Response to Question
> Analysis of convergence rate for BO has been studied in [1,2]. For VAE-based Latent Space BO, the coupling of deep neural networks requires more sufficient assumptions to derive theoretical deduction. For instance, T-LBO [3] manages to prove a probabilistic guarantee for a cumulative regret at rate $1/n$ over the assumptions that 1) the smoothness of the objective function, 2) the bounded posterior variance, and 3) the constant zero-mean Gaussian noise.
>
> While establishing a theoretical framework for convergence is challenging, empirical evidence suggests that TSBO exhibits superior sample efficiency and a faster rate of convergence. This observation is consistent across multiple datasets and various initial conditions, suggesting its potential as a robust and effective approach for BO.
>
> ## Reference
>
> [1] Bull, A. D. (2011). Convergence rates of efficient global optimization algorithms. Journal of Machine Learning Research, 12(10).
>
> [2] Wang, Z., Hutter, F., Zoghi, M., Matheson, D., & De Feitas, N. (2016). Bayesian optimization in a billion dimensions via random embeddings. Journal of Artificial Intelligence Research, 55, 361-387.
>
> [3] Grosnit, A., Tutunov, R., Maraval, A. M., Griffiths, R. R., Cowen-Rivers, A. I., Yang, L., ... & Bou-Ammar, H. (2021). High-dimensional Bayesian optimisation with variational autoencoders and deep metric learning. arXiv preprint arXiv:2106.03609.

---

> > ### Comment · Reviewer_QKF7 · 2023-12-05
> >
> > Thank you for your answers which clarify the points raised.
> >
> > The experiments with n_init = 400 tend to show that TSBO is a promising architecture which has better data efficiency.
> >
> > The proofs given as reference do point out the challenge of deriving a theoretical framework. However I still cannot grasp how the Teacher Student structure can perform better than other BO methods.

---

### Meta-Review · Area_Chair_HXNF · 2023-12-23

**Metareview:**

This work introduces an approach to Bayesian Optimization that utilizes a generative model of more abundant auxiliary unlabeled data to improve representation of high-value regions in the surrogate model. Reviewers found the proposed approach and variety of experiments to be quite interesting, but they also raised significant concerns about the method's complexity, the necessity of certain design choices, and how/why each component works.  For example, reviewer tpuF suggests that other more standard semi-supervised approaches could be adopted, which could simplify the approach.  Reviewer r1KM and the meta-reviewer believe that the present manuscript leaves much to be desired when it comes to understanding how the components of the proposed approach improve efficiency.  The discussion of selective regularization is quite interesting, and it would be great to see more of this discussed in a future submission.

**Justification For Why Not Higher Score:**

This paper would benefit from a more major revision due to potentially unnecessary complexity and lack of intuition / understanding behind why the method works.

**Justification For Why Not Lower Score:**

N/A

---

### Decision · Program_Chairs · 2024-01-16

Reject